# ReKep: Spatio-Temporal Reasoning of Relational Keypoint Constraints for Robotic Manipulation

**Wenlong Huang[1], Chen Wang[1*], Yunzhu Li[2*], Ruohan Zhang[1], Li Fei-Fei[1]**
[1]Stanford University    [2]Columbia University
[rekep-robot.github.io](rekep-robot.github.io)

**Abstract:** Representing robotic manipulation tasks as constraints that associate the robot and the environment is a promising way to encode desired robot behaviors. However, it remains unclear how to formulate the constraints such that they are 1) versatile to diverse tasks, 2) free of manual labeling, and 3) optimizable by off-the-shelf solvers to produce robot actions in real-time. In this work, we introduce **Relational Keypoint Constraints (ReKep)**, a visually-grounded representation for constraints in robotic manipulation. Specifically, ReKep is expressed as Python functions mapping a set of 3D keypoints in the environment to a numerical cost. We demonstrate that by representing a manipulation task as a sequence of Relational Keypoint Constraints, we can employ a hierarchical optimization procedure to solve for robot actions (represented by a sequence of end-effector poses in $SE(3)$) with a perception-action loop at a real-time frequency. Furthermore, in order to circumvent the need for manual specification of ReKep for each new task, we devise an automated procedure that leverages large vision models and vision-language models to produce ReKep from free-form language instructions and RGB-D observations. We present system implementations on a wheeled single-arm platform and a stationary dual-arm platform that can perform a large variety of manipulation tasks, featuring multi-stage, in-the-wild, bimanual, and reactive behaviors, all without task-specific data or environment models.

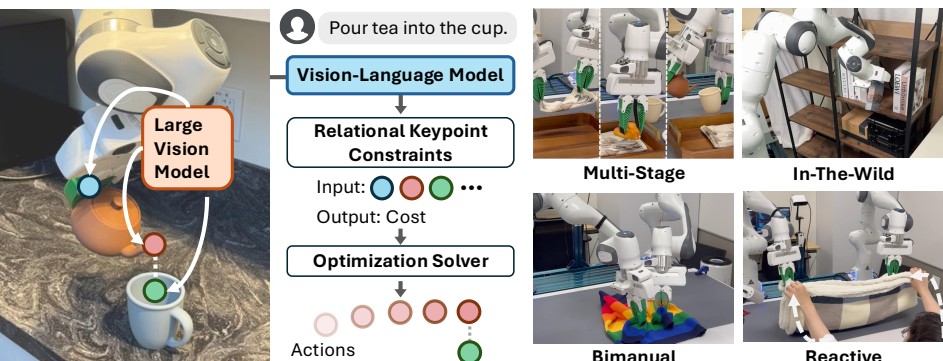

**Figure 1: Relational Keypoint Constraints (ReKep)** specify diverse manipulation behaviors as an optimizable spatio-temporal series of constraint functions operating on semantic keypoints. In the pouring task, one ReKep first constrains the grasping location at the handle of the teapot (*blue*). A subsequent ReKep pulls the teapot spout (*red*) towards the top of the cup opening (*green*) while another ReKep constrains the desired rotation of the teapot by associating the vector formed by the handle (*blue*) and the spout (*red*).

## 1 Introduction

Robotic manipulation involves intricate interactions with objects in the environment, which can often be expressed as constraints in both spatial and temporal domains. Consider the task of pouring tea into a cup in Fig. 1: the robot must grasp *at the handle*, keep the cup *upright* while transporting

---

*Denotes equal contribution. Correspondence to Wenlong Huang <wenlongh@stanford.edu>.

8th Conference on Robot Learning (CoRL 2024), Munich, Germany.

it, *align* the spout with the target container, and then tilt the cup *at the correct angle* to pour. Here, the constraints encode not only the intermediate sub-goals (e.g., align the spout) but also the transitioning behaviors (e.g., keep the cup *upright* in transportation), which collectively dictate the spatial, timing, and other combinatorial requirements of the robot's actions in relation to the environment.

However, effectively formulating these constraints for a large variety of real-world tasks presents significant challenges. While representing constraints using relative poses between robots and objects is a direct and widely-used approach [1], rigid-body transformations do not depict geometric details, require obtaining object models *a priori*, and cannot work on deformable objects. On the other hand, data-driven approaches enable learning constraints directly in visual space [2, 3]. While more flexible, it remains unclear how to effectively collect training data as the number of constraints grows combinatorially in terms of objects and tasks. Therefore, we ask the question: how can we represent constraints in manipulation that are 1) widely applicable: adaptable to tasks that require multi-stage, in-the-wild, bimanual, and reactive behaviors, 2) scalably obtainable: have the potential to be fully automated through the advances in foundation models, and 3) real-time optimizable: can be efficiently solved by off-the-shelf solvers to produce complex manipulation behaviors?

In this work, we propose **Relational Keypoint Constraints (ReKep)**. Specifically, ReKep represents constraints as Python functions that map a set of keypoints to a numerical cost, where each keypoint is a task-specific and semantically meaningful 3D point in the scene. Each function is composed of (potentially nonlinear) arithmetic operations on the keypoints and encodes a desired "relation" between them, where the keypoints may belong to different entities in the environment, such as the robot arms, object parts, and other agents. While each keypoint only consists of its 3D Cartesian coordinates in the world frame, multiple keypoints can collectively specify lines, surfaces, and/or 3D rotations if rigidity between keypoints is enforced. We study ReKep in the context of the sequential manipulation problem, where each task involves multiple stages that have spatio-temporal dependencies (e.g., "grasping", "aligning", and "pouring" in the aforementioned example).

While constraints are typically defined manually per task [4], we demonstrate the specific form of ReKep possesses a unique advantage in that they can be automated by pre-trained large vision models (LVM) [5] and vision-language models (VLM) [6], enabling *in-the-wild* specification of ReKep from RGB-D observations and free-form language instructions. Specifically, we leverage LVM to propose fine-grained and semantically meaningful keypoints in the scene and VLM to write the constraints as Python functions from visual input overlaid with proposed keypoints. This process can be interpreted as grounding fine-grained spatial relations, often those not easily specified with natural language, in an output modality supported by VLM (code) using visual referral expressions.

With the generated constraints, off-the-shelf solvers can be used to produce robot actions by re-evaluating the constraints based on tracked keypoints. Inspired by [7], we employ a hierarchical optimization procedure to first solve a set of waypoints as sub-goals (represented as $SE(3)$ end-effector poses) and then solve the receding-horizon control problem to obtain a dense sequence of actions to achieve each sub-goal. With appropriate instantiation of the problem, we demonstrate that it can be reliably solved at approximately 10 Hz for the tasks considered in this work.

Our contributions are summarized as follows: 1) We formulate manipulation tasks as a hierarchical optimization problem with Relational Keypoint Constraints; 2) We devise a pipeline to automatically specify keypoints and constraints using large vision models and vision-language models; 3) We present system implementations on two real-robot platforms that take as input a language instruction and RGB-D observations, and produce multi-stage, in-the-wild, bimanual, and reactive behaviors for a large variety of manipulation tasks, all without task-specific data or environment models.

## 2  Related Works

**Structural Representations for Manipulation.** Structural representations determine the orchestration of different modules in a manipulation system and yield different implications on the capabilities, assumptions, efficiency, and effectiveness of the system. Rigid-body poses are most commonly

used given well-understood rigid-body motions in free space and their efficiency at modeling long-range dependencies of objects [1, 8–18]. However, since it often requires both geometry and dynamics of environment to be modeled beforehand, various works have studied structural representations using data-driven methods, such as learning object-centric representation [19–34], particle-based dynamics [35–41], and keypoints or descriptors [3, 4, 42–54]. Among them, keypoints have shown great promises given their interpretability, efficiency, generalization to instance variations [4], and ability to model both rigid bodies and deformable objects. However, manual annotation is required per task, thus lacking scalability in open-world settings, which we aim to address in this work.

**Constrained Optimization in Manipulation.** Constraints are often used to impose desired behaviors on robots. Motion planning algorithms use geometric constraints to compute feasible trajectories that avoid obstacles and achieve goals [55–60]. Contact constraints can be used to plan forceful or contact-rich behaviors [61–74]. For sequential manipulation tasks, task and motion planning (TAMP) [1, 8, 13] is a widely used framework, often formulated as constraint satisfaction problems [11, 75–80] with continuous geometric problems as subroutines. Logic-Geometric Programming [81–86] alternatively formulates a nonlinear constrained program over the entire state trajectory, taking into account both logical and geometric constraints. Constraints can either be manually written or learned from data in the form of manifolds [87], feasibility model [86, 88], or signed-distance fields [2, 89]. Inspired by [7], we formulate sequential manipulation tasks as an integrated *continuous* mathematical program that is repeatedly solved in a receding-horizon fashion, with the key difference being that the constraints are synthesized by foundation models.

**Foundation Models for Robotics.** Leveraging foundation models for robotics is an active area of research. We refer readers to [90–93] for overview and recent applications. Here, we focus on VLMs that are capable of incorporating visual inputs [6, 94–98] for manipulation. However, despite showing promises for open-world planning and goal specification [99–113], the caption-guided pre-training scheme of VLMs often limits visual details that can be retained about the image [114–117]. Self-supervised vision models (e.g., DINO [5, 118]), on the other hand, provide fine-grained pixel-level features useful for various vision and robotic tasks [31, 119–124], but there lack effective ways for interpreting open-world semantics pivotal for cross-task generalization. In this work, we leverage their complementary strengths by using DINOv2 [5] for fine-grained keypoint proposal and using GPT-4o [6] for its visual reasoning capability in a supported output modality (code). Similar forms of visual prompting techniques are also explored in concurrent works [99–101, 112, 125]. In this work, we demonstrate ReKep possesses the unique advantages of performing challenging 6-12 DoF tasks, integrated high-level reasoning for reactive replanning, high-frequency closed-loop execution, and generating black-box constraints via visual prompting. More discussion in Appendix A.10.

## 3 Method

Herein we discuss: **(1)** What are Relational Keypoint Constraints (Sec. 3.1)? **(2)** How to formulate manipulation as a constrained optimization problem with ReKep (Sec. 3.2)? **(3)** What is our algorithmic instantiation that can efficiently solve the optimization in real-time (Sec. 3.3)? **(4)** How to automatically obtain ReKep from RGB-D observations and language instructions (Sec. 3.4)?

### 3.1 Relational Keypoint Constraints (ReKep)

Herein we define a single instance of ReKep. For clarity, we assume that a set of $K$ keypoints have been specified (discussed later in Sec. 3.4). Concretely, each keypoint $k_i \in \mathbb{R}^3$ refers to a 3D point on the scene surface with Cartesian coordinates, which is dependent on the task semantics and the environment (e.g., grasp point on the handle, spout).

A single instance of ReKep is a function $f : \mathbb{R}^{K \times 3} \to \mathbb{R}$ that maps an array of keypoints, denoted as $\boldsymbol{k}$, to an unbounded cost, where $f(\boldsymbol{k}) \leq 0$ indicates the constraint is satisfied. The function $f$ is implemented as a stateless Python function, containing NumPy [126] operations on keypoints, which may be nonlinear and nonconvex. In essence, one instance of ReKep encodes one desired spatial relation between keypoints, which may belong to robot arm(s), object parts, and other agents.

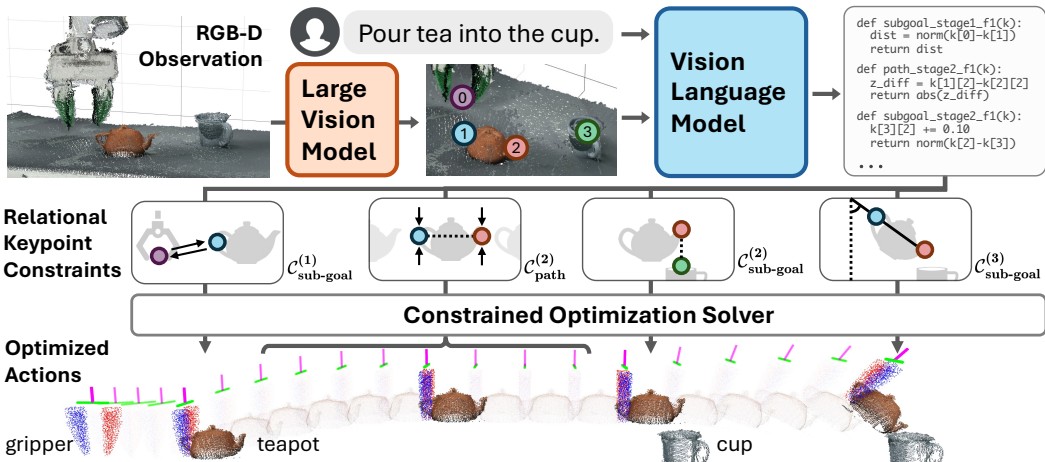

**Figure 2: Overview of ReKep**. DINOv2 [5] first proposes keypoints in the scene, which are overlaid on the original RGB image. The image and an instruction are fed into GPT-4o [6] to generate a series of ReKep constraints as python programs that specify desired relations between keypoints at different task stages ($\mathcal{C}^{(i)}_{\text{sub-goal}}$) and any requirement on transitioning behaviors within stages ($\mathcal{C}^i_{\text{path}}$). Finally, dense sequence of end-effector actions are obtained by solving the optimization problem subject to generated constraints.

However, a manipulation task typically involves multiple spatial relations and may have multiple temporally dependent stages where each stage entails different spatial relations. To this end, we decompose a task into $N$ stages and use ReKep to specify two kinds of constraints for each stage $i \in \{1, \ldots, N\}$: a set of sub-goal constraints $\mathcal{C}^{(i)}_{\text{sub-goal}} = \{f^{(i)}_{\text{sub-goal},1}(\boldsymbol{k}), \ldots, f^{(i)}_{\text{sub-goal},n}(\boldsymbol{k})\}$ and a set of path constraints $\mathcal{C}^{(i)}_{\text{path}} = \{f^{(i)}_{\text{path},1}(\boldsymbol{k}), \ldots, f^{(i)}_{\text{path},m}(\boldsymbol{k})\}$, where $f^{(i)}_{\text{sub-goal}}$ encodes one keypoint relation to be achieved *at the end of stage* $i$, and $f^{(i)}_{\text{path}}$ encodes one keypoint relation to be satisfied for every state *within stage* $i$. Consider the pouring task in Fig. 2, which consists of three stages: grasp, align, and pour. The stage-1 sub-goal constraint pulls the end-effector towards the teapot handle. Then stage-2 sub-goal constraint specifies that the teapot spout needs to be on top of the cup opening. Additionally, stage-2 path constraint ensures the teapot stays upright to avoid spillage when transported. Finally, the stage-3 sub-goal constraint specifies the desired pouring angle.

## 3.2 Manipulation Tasks as Constrained Optimization with ReKep

Using ReKep as a general tool to represent constraints, we adopt the formulation in [7] and show how a manipulation task can be formulated as a constrained optimization problem involving $\mathcal{C}^{(i)}_{\text{sub-goal}}$ and $\mathcal{C}^{(i)}_{\text{path}}$. We denote the end-effector pose as $\mathbf{e} \in SE(3)$. To perform the manipulation task, we aim to obtain the overall discrete-time trajectory $\mathbf{e}_{1:T}$ by formulating the control problem as follows:

$$\underset{\mathbf{e}_{1:T}, g_{1:N}}{\arg\min} \sum_{i=1}^{N} \left[ \lambda^{(i)}_{\text{sub-goal}}(\mathbf{e}_{g_i}) + \sum_{t=g_{i-1}}^{g_i} \lambda^{(i)}_{\text{path}}(\mathbf{e}_t) \right] \text{ s.t. } \begin{cases} \mathbf{e}_1 = \mathbf{e}_{\text{init}}, \ g_0 = 1, \ 0 < g_i < g_{i+1} \\ f(\boldsymbol{k}_{g_i}) \leq 0, \ \forall f \in \mathcal{C}^{(i)}_{\text{sub-goal}} \\ f(\boldsymbol{k}_t) \leq 0, \ \forall f \in \mathcal{C}^{(i)}_{\text{path}}, \ t = g_{i-1}, \ldots, g_i \\ \boldsymbol{k}_{t+1} = h(\boldsymbol{k}_t, \mathbf{e}_t), \ t = 1, \ldots, T-1 \end{cases}$$

(1)

where $\mathbf{e}_t$ denotes end-effector pose at time $t$, $g_i \in \{1, \ldots, T\}$ are the timings of the transition from stage $i$ to $i + 1$ which are also auxiliary decision variables, $\boldsymbol{k}_t$ is the array of keypoint positions at time $t$, $h$ is a forward model of keypoints, and $\lambda^{(i)}_{\text{sub-goal}}$ and $\lambda^{(i)}_{\text{path}}$ are auxiliary cost functions (e.g., collision avoidance) for the sub-goal and path problems respectively. Namely, for each stage $i$, the optimization shall find an end-effector pose as next sub-goal, along with its timing, and a sequence of poses $\mathbf{e}_{g_{i-1}:g_i}$ that achieves the sub-goal, subject to the given set of ReKep constraints and auxiliary costs. This formulation can be considered as direct shooting in trajectory optimization [127].

## 3.3 Decomposition and Algorithmic Instantiation

To solve Eq. 1 in real-time, we employ a decomposition of the full problem and only optimize for the immediate next sub-goal and the corresponding path to reach the sub-goal (pseudo-code in Algorithm 1). All optimization problems are implemented and solved using SciPy [128] with decision variables normalized to $[0, 1]$. They are initially solved with Dual Annealing [129] with SLSQP [130] as local optimizer (around 1 second) and subsequently solved with only local optimizer based on the previous solution at approximately 10 Hz.

**The Sub-Goal Problem**: We first solve the sub-goal problem to obtain $\mathbf{e}_{g_i}$ for the current stage $i$:

$$\underset{\mathbf{e}_{g_i}}{\arg\min} \ \lambda_{\text{sub-goal}}^{(i)}(\mathbf{e}_{g_i}) \quad \text{s.t.} \quad f(\boldsymbol{k}_{g_i}) \leq 0, \ \forall f \in \mathcal{C}_{\text{sub-goal}}^{(i)} \tag{2}$$

where $\lambda_{\text{sub-goal}}$ subsumes auxiliary control costs: scene collision avoidance, reachability, pose regularization, solution consistency, and self-collision for bimanual setup (details in A.8). Namely, Eq. 2 attempts to find a sub-goal that satisfies $\mathcal{C}_{\text{sub-goal}}^{i}$ while minimizing the auxiliary costs. If a stage is concerned with grasping, a grasp metric is also included. In this work, we use AnyGrasp [131][1].

**The Path Problem**: After obtaining sub-goal $\mathbf{e}_{g_i}$, we solve for a trajectory $\mathbf{e}_{t:g_i}$ starting from current end-effector pose $\mathbf{e}_t$ to the sub-goal $\mathbf{e}_{g_i}$:

$$\underset{\mathbf{e}_{t:g_i}, g_i}{\arg\min} \ \lambda_{\text{path}}^{(i)}(\mathbf{e}_{t:g_i}) \quad \text{s.t.} \quad f(\boldsymbol{k}_{\hat{t}}) \leq 0, \quad \forall f \in \mathcal{C}_{\text{path}}^{(i)}, \quad \hat{t} = t, \dots, g_i \tag{3}$$

where $\lambda_{\text{path}}$ subsumes the following auxiliary control costs: scene collision avoidance, reachability, path length, solution consistency, and self-collision in the case of bimanual setup (details in A.9). If the distance to the sub-goal $\mathbf{e}_{g_i}$ is within a small tolerance $\epsilon$, we progress to the next stage $i + 1$.

**Backtracking**: Although the sub-problems can be solved at a real-time frequency to react to external disturbances within a stage, it is imperative that the system can replan across stages if any sub-goal constraint from the last stage no longer holds (e.g., cup taken out of the gripper in the pouring task). Specifically, in every control loop, we check for violation of $\mathcal{C}_{\text{path}}^{(i)}$. If one is found, we iteratively backtrack to a previous stage $j$ such that $\mathcal{C}_{\text{path}}^{(j)}$ is satisfied.

**Forward Models for Keypoints**: To solve Eq. 2 and Eq. 3, one must utilize a forward model $h$ that estimates $\Delta \boldsymbol{k}$ from $\Delta \mathbf{e}$ in the optimization process. As in prior work [4], we make the rigidity assumption between the end-effector and the "grasped keypoints" (a rigid group of keypoints that belong to the same object or part; obtained from the segmentation model as described in Sec. 3.4.). Namely, given a change in the end-effector pose $\Delta \mathbf{e}$, we can calculate the change in keypoint positions by applying the same relative rigid transformation $\boldsymbol{k}'[\text{grasped}] = \mathbf{T}_{\Delta \mathbf{e}} \cdot \boldsymbol{k}[\text{grasped}]$, while assuming other keypoints stay static. We note that this is a "local" assumption in that it is only assumed to hold for the short duration (0.1s) that the problem is solved. Actual keypoint positions are tracked using visual input at 20 Hz and used in every new problem. For more challenging scenarios (e.g., dynamic or contact-rich tasks), a learned or physics-based model may be used.

## 3.4 Keypoint Proposal and ReKep Generation

To enable the system to perform tasks in-the-wild given a free-form task instruction, we devise a pipeline using large vision models and vision-language models for keypoint proposal and ReKep generation, which are respectively discussed as follows:

**Keypoint Proposal**: Given an RGB image $\mathbb{R}^{h \times w \times 3}$, we first extract the patch-wise features $\mathbf{F}_{\text{patch}} \in \mathbb{R}^{h' \times w' \times d}$ from DINOv2 [5]. Then we perform bilinear interpolation to upsample the features to the original image size, $\mathbf{F}_{\text{interp}} \in \mathbb{R}^{h \times w \times d}$. To ensure the proposal covers all relevant objects in the scene, we extract all masks $\mathbf{M} = \{\mathbf{m}_1, \mathbf{m}_2, \dots, \mathbf{m}_n\}$ in the scene using Segment Anything

---

[1]Since AnyGrasp is a grasp detector instead of a metric and is computationally expensive to run in optimization loops, we always return the grasp closest to a specified "grasp keypoint" by exploiting the fact that ReKep related to grasping always associates a dummy keypoint on the end-effector and one actual keypoint.

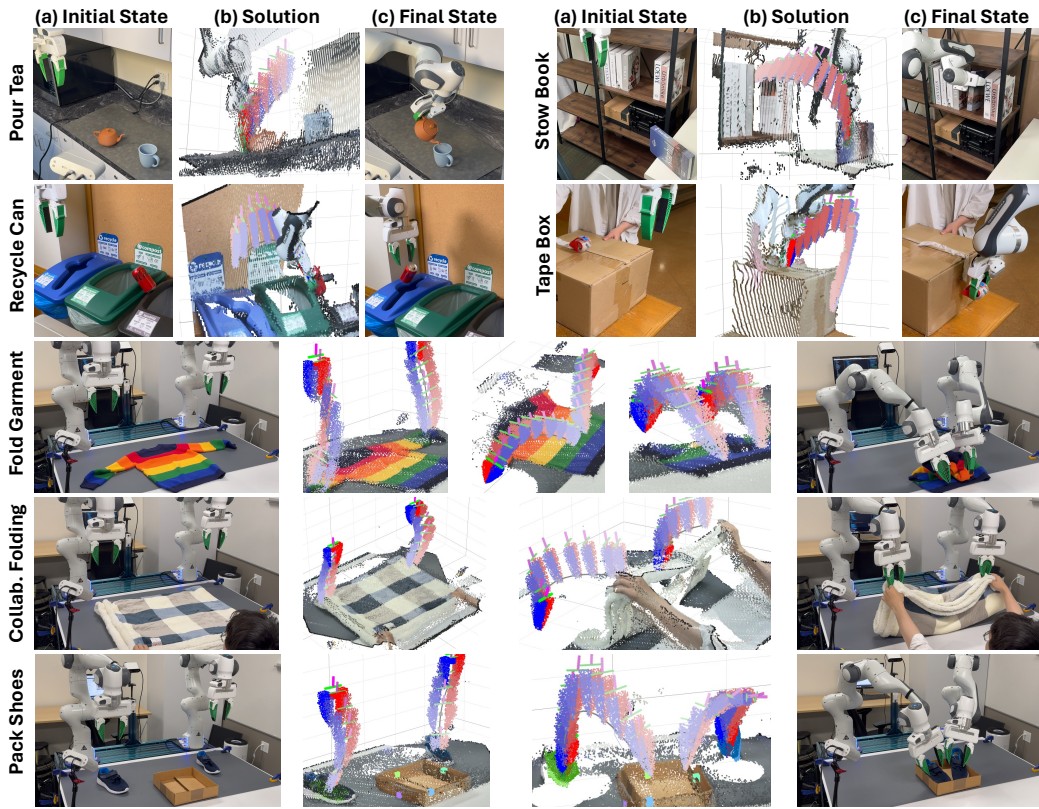

**Figure 3: Experiment tasks and visualization of optimization results.** Seven tasks are designed to validate different aspects of our system, including in-the-wild specification with commonsense knowledge, multi-stage tasks with spatio-temporal dependencies, bimanual coordination with geometric awareness, and reactiveness when collaborating with humans and under disturbances.

(SAM) [132]. For each mask $j$, we cluster the masked features $\mathbf{F}_{\text{interp}}[\mathbf{m}_j]$ using $k$-means with $k = 5$ with a cosine-similarity metric. The centroids of the clusters are used as keypoint candidates, which are projected to a world coordinate $\mathbb{R}^3$ using a calibrated RGB-D camera. Candidates that are within 8cm of others are filtered out. Overall, we find that this procedure is adept at identifying a large percentage of fine-grained and semantically meaningful regions of objects.

**ReKep Generation**: After obtaining the keypoint candidates, we overlay them on the original RGB image with numerical marks. Coupled with the language instruction of the task, we then use visual prompting to query GPT-4o [6] to generate the number of required stages and the corresponding sub-goal constraints $\mathcal{C}_{\text{sub-goal}}^{(i)}$ and path constraints $\mathcal{C}_{\text{path}}^{(i)}$ for each stage $i$ (prompts are in A.6). Notably, the functions do not directly manipulate the numerical values of the keypoint positions. Rather, we exploit the strength of VLM to specify spatial relations as *arithmetic operations*, such as L2 distance or dot product between keypoints, that are only instantiated when invoked with actual keypoint positions tracked by a specialized 3D tracker. Furthermore, an important advantage of using arithmetic operations on a set of keypoint positions is that it can specify 3D rotations in full $SO(3)$ when sufficient points are provided and rigidity between relevant points is enforced, but this is done only when needed depending on task semantic. This enables VLM to reason about 3D rotations with arithmetic operations *in 3D Cartesian space*, effectively circumventing the need for dealing with alternative 3D rotation representation and the need for performing numerical computation.

## 4   Experiments

We aim to answer the following research questions: **(1)** How well does our framework automatically formulate and synthesize manipulation behaviors (Sec. 4.1)? **(2)** Can our system generalize to novel

objects and manipulation strategies (Sec. 4.2)? **(3)** How do the individual components contribute to the failure cases of the system (Sec. 4.3)? We validate ReKep on two real robot platforms: a wheeled single-arm platform, and a stationary dual-arm platform (Figure. 3). Additional implementation details can be found in Appendix, including keypoint proposal (A.5), VLM querying (A.6), point trackers (A.7), sub-goal solver (A.8), and path solver (A.9).

## 4.1 ReKep for In-the-Wild and Bimanual Manipulation

**Tasks.** We purposefully select a set of tasks (shown in Fig. 3) with the goal of examining the multi-stage (**m**), in-the-wild (**w**), bimanual (**b**), and reactive (**r**) behaviors of the system. The tasks and their features are *Pour Tea* (**m**, **w**, **r**), *Stow Book* (**w**), *Recycle Can* (**w**), *Tape Box* (**w**, **r**), *Fold Garment* (**b**), *Pack Shoes* (**b**), and *Collaborative Folding* (**b**, **r**). We further evaluate three of the tasks under external disturbances (denoted as "Dist.") by changing poses of task objects during execution.

**Metric and Baselines.** Each setting has 10 trials, in which object poses are randomized. Success rate is reported in Tab. 1. We compare to VoxPoser [106] as a baseline. We evaluate two variants of the system: "Auto" uses foundation models to automatically generate ReKep, and "Annotated (Annot.)" uses human-annotated ReKep.

**Results.** Compared to baselines, ReKep can effectively handle core challenges of each task. For example, it can formulate correct temporal dependency in multi-stage tasks (e.g., spout needs to be aligned with the cup before pouring), leverage commonsense knowledge (e.g., coke cans should be recycled), and construct coordination behaviors in both bimanual settings (e.g., folding left sleeve and right sleeve simultaneously) and human-robot collaboration setting (e.g., folding a large blanket by aligning the four corners together with human). Coupled with an optimization framework, it can also generate kinematically challenging behaviors in confined spaces in the *Stow Book* task and find a feasible solution that densely fits two shoes within a small volume in the *Pack Shoes* task. Since the keypoints are tracked at a high frequency, the system can react to external disturbances and replan both within stage and across stages. Despite promising results, we also identify several limitations which are discussed in Sec. 5.

|  | | ReKep | |
| --- | --- | --- | --- |
| **Task** | **VoxPoser** | Auto | Annot. |
| **Mobile Arm** | | | |
| Pour Tea | 0/10 | 3/10 | 8/10 |
| Recycle Can | 3/10 | 6/10 | 8/10 |
| Stow Book | 0/10 | 3/10 | 6/10 |
| Tape Box | 4/10 | 7/10 | 8/10 |
| **Dual-Arm** | | | |
| Fold Garment | 0/10 | 5/10 | 6/10 |
| Pack Shoes | 0/10 | 3/10 | 5/10 |
| Collab. Folding | 0/10 | 4/10 | 7/10 |
| **Total (%)** | 10.0% | 44.3% | 68.6% |

**Table 1:** Success rate on wheeled single-arm and stationary bimanual platforms.

|  | | ReKep | |
| --- | --- | --- | --- |
| **Task (Dist.)** | **VoxPoser** | Auto | Annot. |
| Pour Tea | 0/10 | 2/10 | 4/10 |
| Tape Box | 2/10 | 3/10 | 5/10 |
| Collab. Folding | 0/10 | 3/10 | 5/10 |
| **Total (%)** | 6.7% | 26.7% | 46.7% |

**Table 2:** Success rate under external disturbances across both robot platforms.

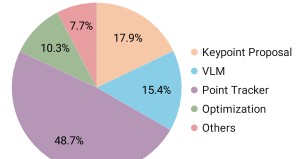

**Figure 4:** System error breakdown.

## 4.2 Generalization in Manipulation Strategies

**Tasks.** We systematically evaluate how novel manipulation strategies can be formulated by focusing on a single task, garment folding, but with 8 unique categories of garments, each demanding a unique way of folding and requiring both geometrical and commonsense reasoning. Evaluation is done on the bimanual platform, presenting additional challenges in bimanual coordination.

**Metric.** We use GPT-4o with a prompt containing only generic instructions with no in-context examples. "Strategy Success" measures whether generated ReKep is feasible, which tests both the keypoint proposal module and the VLM, and "Execution Success" measures system success rate given feasible strategies for each clothing. Each is measured with 10 trials.

**Results.** Interestingly, we observe drastically different strategies across categories, many of which are aligned with how humans might fold each garment. For example, it can recognize that two

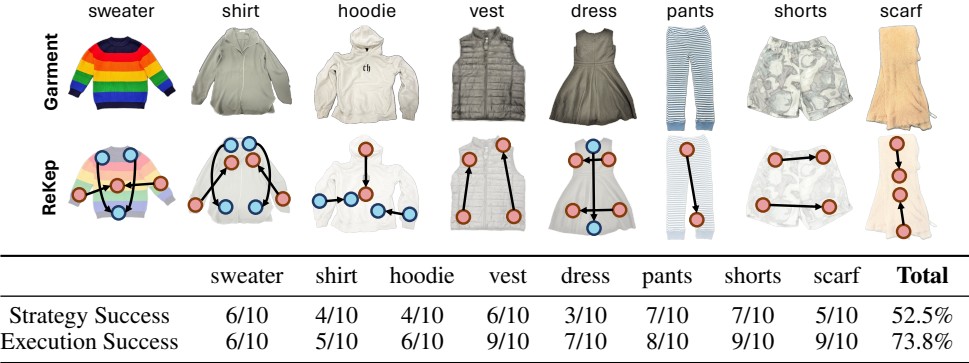

| | sweater | shirt | hoodie | vest | dress | pants | shorts | scarf | **Total** |
|---|---|---|---|---|---|---|---|---|---|
| Strategy Success | 6/10 | 4/10 | 4/10 | 6/10 | 3/10 | 7/10 | 7/10 | 5/10 | 52.5% |
| Execution Success | 6/10 | 5/10 | 6/10 | 9/10 | 7/10 | 8/10 | 9/10 | 9/10 | 73.8% |

**Figure 5:** Novel *bimanual* strategies of ReKep for folding different categories of garments and their success rates. Since ReKep in this task always associates two points at a time, two keypoints are connected by an arrow if they need to be aligned. The coloring of the keypoints denotes the order. In the sweater task, two sleeves are first folded simultaneously with two arms, and then the two arms grasp the crew neck to align to the bottom.

sleeves often are folded together, prior to fully folding the clothes. In cases where using two arms is unnecessary, akin to how humans fold clothes, only one arm is being used. However, we do observe that the VLM may miss certain steps to complete the folding as the operator expected, but we recognize that this is inherently an open-ended problem often based on one's preferences.

### 4.3 System Error Breakdown

The modular design of the framework entails an advantage for analyzing system errors due to its interpretability. In this section, we perform an empirical investigation by manually inspecting the failure cases of the experiments reported in Tab. 1, which is then used to calculate the likelihood of a module causing an error while accounting for their temporal dependencies in the pipeline. Results are reported in Fig. 4. Among the different modules, the point tracker produces the largest portion of errors, as frequent and intermittent occlusion poses significant challenges for accurate tracking. Keypoint proposal and VLM also produce considerable portions of errors, where common cases include the proposal module missing certain keypoints and the VLM referring to incorrect keypoints. The optimization module, on the other hand, does not contribute as much to the failures despite given limited time budget, since there often exist many possible solutions for each problem. Other modules, such as segmentation, 3D reconstruction, and low-level controller, also contribute to some failure cases, but they are relatively insignificant compared to other modules.

## 5 Conclusion & Limitations

In this work, we presented Relational Keypoint Constraints (ReKep), a structural task representation using constraints that operates on semantic keypoints to specify desired relations between robot arms, object (parts), and other agents in the environment. Coupled with point trackers, we demonstrate that ReKep constraints can be repeatedly and efficiently solved in a hierarchical optimization framework to act as a closed-loop policy that runs at a real-time frequency. We also demonstrate the unique advantage of ReKep in that it can be automatically synthesized by large vision models and vision-language models. Results are shown on two robot platforms and on a variety of tasks featuring multi-stage, in-the-wild, bimanual, and reactive behaviors, all without task-specific data, additional training, or environment models. Despite the promises, several limitations remained. First, the optimization framework relies on a forward model of keypoints based on rigidity assumption, albeit a high-frequency feedback loop that relaxes the accuracy requirement of the model. Second, ReKep relies on accurate point tracking to correctly optimize actions in closed-loop, which is itself a challenging 3D vision task due to heavy intermittent occlusions. Lastly, the current formulation assumes a fixed sequence of stages (i.e., skeletons) for each task. Replanning with different skeletons requires running keypoint proposal and VLM at a high-frequency, which poses considerable computational challenges. An extended discussion of limitations can be found in Appendix A.11.

**Acknowledgments**

This work is partially supported by Stanford Institute for Human-Centered Artificial Intelligence, ONR MURI N00014-21-1-2801, and Schmidt Sciences. Ruohan Zhang is partially supported by Wu Tsai Human Performance Alliance Fellowship. The bimanual hardware is partially supported by Stanford TML. We would like to thank the anonymous reviewers, Albert Wu, Yifan Hou, Adrien Gaidon, Adam Harley, Christopher Agia, Edward Schmerling, Marco Pavone, Yunfan Jiang, Yixuan Wang, Sirui Chen, Chengshu Li, Josiah Wong, Wensi Ai, Weiyu Liu, Mengdi Xu, Yihe Tang, Chelsea Ye, Mijiu Mili, and the members of the Stanford Vision and Learning Lab for fruitful discussions, helps on experiments, and support.

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

# A Appendix

## A.1 Pseudo-code for Sequential Manipulation with Relational Keypoint Constraints

---

**Algorithm 1** Relational Keypoint Constraints for Sequential Manipulation

---

1: Initialize current stage $i \leftarrow 1$, and current time $t \leftarrow 1$
2: **while** $i \leq N$ **do**
3:     **if** $\exists f \in \mathcal{C}_{\text{path}}^{(i)}$ s.t. $f(\boldsymbol{k}_t) > 0$ **then**
4:         $i \leftarrow i - 1$
5:         **continue**
6:     **end if**
7:     **if** distance$(\mathbf{e}_t, \mathbf{e}_{g_i}) < \epsilon$ **then**
8:         $i \leftarrow i + 1$
9:         **continue**
10:    **end if**
11:    Solve sub-goal problem for stage $i$ to obtain $\mathbf{e}_{g_i}$ (Eq. 2)
12:    Solve path problem for stage $i$ to obtain $\mathbf{e}_{t:g_i}$ (Eq. 3)
13:    Execute the next $m$ actions $\mathbf{e}_{t+1:t+m}$
14:    $t \leftarrow t + m + 1$
15: **end while**

---

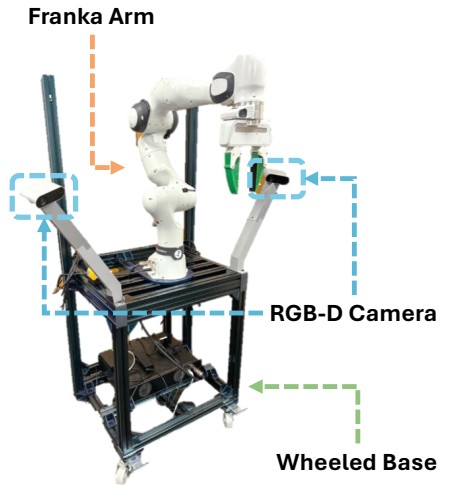

**Figure 6:** Wheeled Single-Arm Platform.

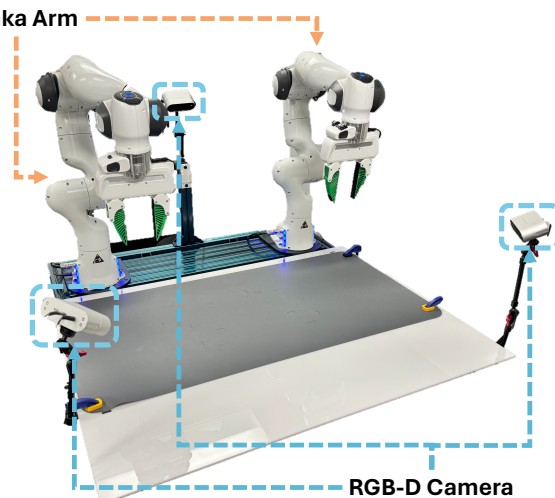

**Figure 7:** Stationary Dual-Arm Platform.

## A.2 Wheeled Single-Arm Platform

One of our investigated platform is a Franka arm mounted on a wheeled base built with Vention frames (shown in Figure 6). Note that the base does not have motors and thus cannot move autonomously, but its mobility nevertheless allows us to investigate the proposed method outside of lab environments.

Since our pipeline produces a sequence of 6-DoF end-effector poses, we use position control in all experiments, which is running at a fixed frequency of 20 Hz. Specifically, once the robot is given a target end-effector pose in the world frame, we first clip the pose to the pre-defined workspace. Then we linearly interpolate from the current pose to the target pose with a step size of 5mm for position and 1 degree for rotation. To move to each interpolated pose, we first calculate inverse kinematics to obtain the target joint positions based on current joint positions (IK solver from PyBullet [133]). Then we use the joint impedance controller from Deoxys [134] to reach to the target joint positions.

Two RGB-D cameras, Orbbec Femto Bolt, are mounted on each side of the robot facing the workspace center. The cameras capture RGB images and point clouds at a fixed frequency of 20 Hz.

### A.3 Stationary Dual-Arm Platform

We also investigate the method on a stationary dual-arm platform consisting of two Franka arms mounted in front of a tabletop workspace (shown in Figure 7). We share the same controller as the wheeled single-arm platform with the exception that the two arms are controlled simultaneously at 20 Hz. Specifically, our pipeline jointly solves two 6-DoF end-effector pose sequences, which are sent to the low-level controller together. The controller subsequently calculates IK for both arms and moves the arms using joint impedance control.

Three RGB-D cameras, Orbbec Femto Bolt, are mounted on this platform. Two cameras are mounted on the left and right sides and one camera is mounted in the back. The cameras similarly capture RGB images and point clouds at a fixed frequency of 20 Hz.

### A.4 Evaluation Details

Below we discuss the evaluation details for the experiments reported in Section 4.1 and Section 4.2.

#### A.4.1 Details for In-the-Wild and Bimanual Manipulation (Section 4.1)

For each task, 10 initial different configurations of objects are selected, which cover the full workspace but are manually verified to ensure they are kinematically feasible for the robot. For each trial, a human operator restores the scene to the corresponding configuration and initiates the system. Due to the challenge of developing automatic success criteria for the diverse set of objects and environments investigated in this work, success rates are measured by the operator with the criterion reported under each task description below. For experiments involving external disturbances, the set of disturbances for all trials is pre-selected, and one disturbance is applied to each trial. Specifically, the disturbance is introduced by a human operator using hands to change the object's pose. Collision checking is disabled for all tasks involving deformable objects.

**Pour Tea**: The environment consists of a teapot and a cup placed on a counter table in a kitchen setting. The task involves three stages: grasping the handle, aligning the teapot to the top of the cup, and pouring the tea into the cup. The success criterion requires that the teapot remains upright until the pouring stage, and at the end, the spout must be aligned and tilted on top of the cup opening.

**Recycle Can**: The environment includes one of three types of cans (Coke, Zero Coke, Zero Sprite), a recycle bin with a narrow opening (such that the cans may only go in when they are upright), a landfill bin, and a compost bin, all situated inside an office building. The task involves two stages: grasping the can and reorienting it on top of the recycle bin before dropping it. The success criterion is that the can is successfully thrown into the bin.

**Stow Book**: The environment consists of a target book placed on a side table and a real-size bookshelf with a 15cm opening among the placed books, all inside an office environment. The task involves two stages: grasping the target book on the side and stowing it inside the opening in the shelf. The success criterion is that the target book is placed steadily after the robot releases the gripper, and the robot must not bump into the shelf or other placed books.

**Tape Box**: The environment includes a cardboard box, a packaging tape with a dispenser sitting on top of the box that already has one side taped, and a human user collaborating with the robot. The tape has already been unrolled to be enough for taping because unrolling typically requires a large force that exceeds the limit of the robot arm. The task involves two stages: while a human operator is squeezing the box, the robot needs to grasp the tape and align it to the correct side to complete the taping. The success criterion is that the tape must end up in the correct position such that it is aligned with the seam.

**(Bimanual) Fold Garment**: The environment consists of a sweater placed flat close to the workspace center, with small deformations on the sleeves, neck, and bottom. The task typically requires four stages: grasping both sleeves, folding them to the middle, grasping the neck, and folding it to the bottom. The success criterion does not enforce consistent stages; as long as the sweater is folded such that it occupies at most half of the original surface size, it is regarded as a success.

**(Bimanual) Pack Shoes**: The environment includes an empty shoe box placed close to the workspace center, with two shoes placed on opposite sides of the box in random poses. The task involves two stages: grasping the shoes simultaneously and placing them in the shoe box. The success criterion does not enforce consistent stages; as long as the shoes are placed into the box without being stacked together or causing bimanual self-collision, it is considered successful.

**(Bimanual) Collaborative Folding**: The environment consists of a large blanket (pre-folded to an appropriate size that occupies about 70% of the workspace due to its size exceeding the workspace limit) and a human user collaborating with the robot. The task involves two stages: the robot must grasp the two corners of the blanket opposite to the human user, and the second stage is aligning the two corners with the two corners that the human has grasped. The success criterion is that the robot has grasped the correct corners and can align them with the correct human arms (left-left, right-right).

### A.4.2 Details on Baseline Methods

We use VoxPoser [106] as the main baseline method as it makes similar assumptions that no task-specific data or pre-defined motion primitives are required. We adapt VoxPoser to our setting with certain modifications to ensure fairness in comparisons. Specifically, we use the same VLM, GPT-4o [6], that takes the same camera input. We also augment the original prompt from the paper with the prompt used in this work to ensure it has sufficient context. We only use the affordance, rotation, and gripper action value maps and ignore the avoidance and velocity value maps because they are not necessary for our tasks. We also only consider the scenario where the "entity of interest" is the robot end-effector instead of objects in the scene. The latter is tailored for pushing task, which is not being studied in this work. We use OWL-ViT [135] for open-vocabulary object detection, SAM [132] for initial-frame segmentation, and Cutie [136] for mask tracking.

### A.4.3 Details for Generalization in Manipulation Strategies (Section 4.2)

The dual-arm robot is tasked with folding eight different categories of clothing. We use two metrics for evaluation: "Strategy Success" and "Execution Success," where the former evaluates whether keypoints are proposed and constraints are written appropriately, and the latter evaluates the robotic system's execution given successful strategies.

To evaluate "Strategy Success," the garment is initialized close to the center of the workspace. A back-mounted RGB-D camera captures the RGB image. Then, the keypoint proposal module generates keypoint candidates using the captured image, which are then overlaid on top of the original image with numerical marks $\{0, \ldots, K - 1\}$. The overlaid image, along with the same generic prompt, is fed into GPT-4 [6] to generate the ReKep constraints. Since folding garments is itself an open-ended problem without ground-truth strategies, we manually judge if the proposed keypoints and the generated constraints are correct. Note that since the constraints are to be executed by a bimanual robot, and the constraints are almost always connecting (folding) two keypoints such that they are aligned, correctness is measured by whether it is (potentially) executable by the robot without causing self-collision (arms crossing over to opposite sides) and whether the folding strategy can fold the garment to at most half of its original surface area.

To evaluate "Execution Success," we take the generated strategies in the previous section that are marked as successful for each garment and execute the sequence on the dual-arm platform, with a total of 10 trials for each garment. Point tracking is disabled as we observe that our point tracker predicts unstable tracks when the garment is potentially folded many times. Success is measured by whether the garment is folded such that its surface area is at most half of its original surface area.

## A.5 Implementation Details of Keypoint Proposal

Herein we describe how keypoint candidates in a scene are generated. For each platform, we use one of the mounted RGB-D cameras to capture an image of size $h \times w \times 3$, depending on which camera has the best holistic view of the environment, as all the keypoints need to be present in the first frame for the proposed method. Given the captured image, we first use DINOv2 with registers (ViT-S14) [5, 137] to extract the patch-wise features $\mathbf{F}_{\text{patch}} \in \mathbb{R}^{h' \times w' \times d}$. Then we perform bilinear interpolation to upsample the features to the original image size, $\mathbf{F}_{\text{interp}} \in \mathbb{R}^{h \times w \times d}$. To ensure the proposal covers all relevant objects in the scene, we extract all masks $\mathbf{M} = \{\mathbf{m}_1, \mathbf{m}_2, \ldots, \mathbf{m}_n\}$ in the scene using Segment Anything (SAM) [132]. Within each mask $\mathbf{m}_i$, we apply PCA to project the features to three dimensions, $\mathbf{F}_{\text{PCA}} = \text{PCA}(\mathbf{F}_{\text{resized}}[\mathbf{m}_i], 3)$. We find that applying PCA improves the clustering as it often removes details and artifacts related to texture that are not useful for our tasks. For each mask $j$, we cluster the masked features $\mathbf{F}_{\text{interp}}[\mathbf{m}_j]$ using $k$-means with $k = 5$ with the Euclidean distance metric. The median centroids of the clusters are used as keypoint candidates, which are projected to a world coordinate $\mathbb{R}^3$ using a calibrated RGB-D camera. Note that we also store which keypoint candidates originate from the same mask, which is later used as part of the rigidity assumption in the optimization loops described in Sec. 3.3. Candidates outside of the workspace bounds are filtered out. To avoid many points cluttered in a small region, we additionally use Mean Shift [138, 139] (with a bandwidth 8cm) to filter out points that are close to each other. Finally, the centroids are taken as final candidates. Alternatively, one may develop a pipeline using only segmentation models [132, 140], but we leave comparisons to future work.

## A.6 Querying Vision-Language Model

After we obtain the keypoint candidates, they are overlaid on the captured RGB image with numerical marks $\{0, \ldots, K - 1\}$. Then the image and the task instruction are fed into a vision-language model with the prompt described below. The prompt contains only generic instructions with no image-text in-context examples, although a few text-based examples are given to concretely explain the proposed method and the expected output from the model. Note that the majority of the investigated tasks are not discussed in the provided prompt. As a result, the VLM is tasked with generating ReKep constraints by leveraging its internalized world knowledge.

For the experiments conducted in this work, we use GPT-4o [6] as it is one of the latest available models at the time of the experiments. However, due to rapid advancement in this field, the pipeline can directly benefit from newer models that have better vision-language reasoning. Correspondingly, we observe different models exhibit different behaviors when given the same prompt (with the observation that newer models typically require less fine-grained instructions). As a result, instead of developing the best prompt for the suite of tasks in this work, we focus on demonstrating a full-stack pipeline consisting a key component that can be automated and continuously improved by future development.

```
## Instructions
Suppose you are controlling a robot to perform manipulation tasks by writing constraint functions in Python.
    The manipulation task is given as an image of the environment, overlaid with keypoints marked with
    their indices, along with a text instruction. The instruction starts with a parenthesis indicating
    whether the robot has a single arm or is bimanual. For each given task, please perform the following
    steps:
- Determine how many stages are involved in the task. Grasping must be an independent stage. Some examples:
  - "(single-arm) pouring tea from teapot":
    - 3 stages: "grasp teapot", "align teapot with cup opening", and "pour liquid"
  - "(single-arm) put red block on top of blue block":
    - 3 stages: "grasp red block", "align red block on top of blue block", and "release red block"
  - "(bimanual) fold sleeves to the center":
    - 2 stages: "left arm grasps left sleeve and right arm grasps right sleeve" and "both arms fold sleeves to
        the center"
  - "(bimanual) fold a jacket":
    - 3 stages: "left arm grasps left sleeve and right arm grasps right sleeve", "both arms fold sleeves to
        the center", and "grasp the neck with one arm (the other arm stays in place)", and "align the neck
        with the bottom"
- For each stage, write two kinds of constraints, "sub-goal constraints" and "path constraints". The "sub-goal
    constraints" are constraints that must be satisfied **at the end of the stage**, while the "path
    constraints" are constraints that must be satisfied **within the stage**. Some examples:
  - "(single-arm) pouring liquid from teapot":
    - "grasp teapot" stage:
    - sub-goal constraints: "align the end-effector with the teapot handle"
    - path constraints: None
```

```
        - "align teapot with cup opening" stage:
          - sub-goal constraints: "the teapot spout needs to be 10cm above the cup opening"
          - path constraints: "robot is grasping the teapot", and "the teapot must stay upright to avoid spilling"
        - "pour liquid" stage:
          - sub-goal constraints: "the teapot spout needs to be 5cm above the cup opening", "the teapot spout must
                  be tilted to pour liquid"
          - path constraints: "the teapot spout is directly above the cup opening"
    - "(bimanual) fold sleeves to the center":
        - "left arm grasps left sleeve and right arm grasps right sleeve" stage:
          - sub-goal constraints: "left arm grasps left sleeve", "right arm grasps right sleeve"
          - path constraints: None
        - "both arms fold sleeves to the center" stage:
          - sub-goal constraints: "left sleeve aligns with the center", "right sleeve aligns with the center"
          - path constraints: None

Note:
- Each constraint takes a dummy end-effector point and a set of keypoints as input and returns a numerical
      cost, where the constraint is satisfied if the cost is smaller than or equal to zero.
- For each stage, you may write 0 or more sub-goal constraints and 0 or more path constraints.
- Avoid using "if" statements in your constraints.
- Avoid using path constraints when manipulating deformable objects (e.g., clothing, towels).
- You do not need to consider collision avoidance. Focus on what is necessary to complete the task.
- Inputs to the constraints are as follows:
   - `end_effector`: np.array of shape `(3,)` representing the end-effector position.
   - `keypoints`: np.array of shape `(K, 3)` representing the keypoint positions.
- Inside of each function, you may use native Python functions and NumPy functions.
- For grasping stage, you should only write one sub-goal constraint that associates the end-effector with a
      keypoint. No path constraints are needed.
- For non-grasping stage, you should not refer to the end-effector position.
- In order to move a keypoint, its associated object must be grasped in one of the previous stages.
- The robot can only grasp one object at a time.
- Grasping must be an independent stage from other stages.
- You may use two keypoints to form a vector, which can be used to specify a rotation (by specifying the angle
       between the vector and a fixed axis).
- You may use multiple keypoints to specify a surface or volume.
- You may also use the center of multiple keypoints to specify a position.
- A single folding action should consist of two stages: one grasp and one place.

Structure your output in a single python code block as follows for single-arm robot:
```python

# Your explanation of how many stages are involved in the task and what each stage is about.
# ...

num_stages = ?

### stage 1 sub-goal constraints (if any)
def stage1_subgoal_constraint1(end_effector, keypoints):
    """Put your explanation here."""
    ...
    return cost
# Add more sub-goal constraints if needed

### stage 1 path constraints (if any)
def stage1_path_constraint1(end_effector, keypoints):
    """Put your explanation here."""
    ...
    return cost
# Add more path constraints if needed

# repeat for more stages
...
```

Structure your output in a single python code block as follows for bimanual robot:
```python

# Your explanation of how many stages are involved in the task and what each stage is about.
# ...

num_stages = ?

### left-arm stage 1 sub-goal constraints (if any)
def left_stage1_subgoal_constraint1(end_effector, keypoints):
    """Put your explanation here."""
    ...
    return cost

### right-arm stage 1 sub-goal constraints (if any)
def right_stage1_subgoal_constraint1(end_effector, keypoints):
    """Put your explanation here."""
    ...
    return cost
# Add more sub-goal constraints if needed

### left stage 1 path constraints (if any)
def left_stage1_path_constraint1(end_effector, keypoints):
    """Put your explanation here."""
    ...
    return cost
### right stage 1 path constraints (if any)
def right_stage1_path_constraint1(end_effector, keypoints):
```

```
    """Put your explanation here."""
    ...
    return cost
# Add more path constraints if needed

# repeat for more stages
...
...

## Query
Query Task: "[INSTRUCTION]"
Query Image: [IMAGE WITH KEYPOINTS]
```

## A.7   Implementation Details of Point Tracker

We implement a simple point tracker following [121] based on DINOv2 (ViT-S14) [5] that leverages the fact that multiple RGB-D cameras are present and DINOv2 is efficient to run at a real-time frequency.

At initialization, an array of 3D keypoint positions $k \in \mathbb{R}$ are given. We first take the RGB-D captures from each present camera. For each RGB image, we obtain the pixel-wise DINOv2 features following the same procedure in Section A.5 and record their associated 3D world coordinates using calibrated cameras. For each 3D keypoint positions, we aggregate all the features from points that are within 2cm from all the cameras. The mean of the aggregated features is recorded as the reference feature for each keypoint, which is kept fixed throughout the task.

After initialization, at each time step, we similarly obtain the pixel-wise features from DINOv2 from all cameras with their 3D world coordinates. To track the keypoints, we calculate cosine similarity between features across all pixels and the reference features. The top $100$ matches are selected for each keypoint with a cutoff similarity of $0.6$. We then reject outliers for the selected matches by calculating median deviation ($m = 2$). Additionally, as the tracked keypoints may oscillate in a small region, we apply a uniform filter with a window size of $10$ in the end. The entire procedure runs at a fixed frequency of 20 Hz.

Note that the implemented point tracker is a simplification from [121] for real-time tracking. We refer readers to [121] for more comprehensive discussion on using self-supervised vision models, such as DINOv2, for point tracking. Alternatively, more specialized point trackers can be used [141–148].

## A.8   Implementation Details of Sub-Goal Solver

The sub-goal problems are implemented and solved using SciPy [128]. The decision variable is a single end-effector pose (position and Euler angles) in $\mathbb{R}^6$ for single-arm robots and two end-effector poses in $\mathbb{R}^{12}$ for bimanual robot. The bounds for the position terms are the pre-defined workspace bounds, and the bounds for the rotation terms are that the half hemisphere where the end-effector faces down (due to the joint limits of the Franka arm, it is often likely to reach joint limit when an end-effector pose faces up). The decision variables are normalized to $[-1, 1]$ based on the bounds. For the first solving iteration, the initial guess is chosen to be the current end-effector pose. We use sampling-based global optimization Dual Annealing [129] in the first iteration to quickly search the full space, which is followed by a gradient-based local optimizer SLSQP [130] that refines the solution. The full procedure takes around 1 second for this iteration. In subsequent iterations, we use the solution from previous stage and only use local optimizer as it can quickly adjust to small changes. The optimization is cut off with a fixed time budget represented as number of objective function calls to keep the system running at a high frequency.

We discuss the cost terms in the objective function below.

**Constraint Violation**: We implement constraints as cost terms in the optimization problem, where the returned costs by the ReKep functions are multiplied with large weights.

**Scene Collision Avoidance**: We use nvblox [149] with the PyTorch wrapper [58] to compute the ESDF of the scene in a separate node that runs at 20 Hz. The ESDF calculation aggregates the

depth maps from all available cameras and excludes robot arms using cuRobo and any grasped rigid objects (tracked via a masked tracker model Cutie [136]). A collision voxel grid is then calculated using the ESDF and used by other modules in the system. In the sub-goal solver module, we first downsample the gripper points and the grasped object points to have a maximum of 30 points using farthest point sampling. Then we calculate the collision cost using the ESDF voxel grid with linear interpolation with a threshold of 15cm.

**Reachability**: Since our decision variables are end-effector poses, which may not be always reachable by the robot arms, especially in confined spaces, we need to add a cost term that encourages finding solutions with valid joint configurations. Therefore, we solve an IK problem in each iteration of the sub-goal solver using PyBullet [133] and use its residual as a proxy for reachability. We find that this takes around 40% of the time of the full objective function. Alternatively, one may solve the problem in joint space, which would ensure the solution is within the joint limits by enforcing the bounds. We find that this is inefficient with our Python-based implementation as we need to calculate forward kinematics for a magnitude of more times in the path solver, because the constraints are evaluated in the task space. To address this while ensuring efficiency, future works can consider using hardware-accelerated implementations to solve the problems in joint space [58].

**Pose Regularization**: We also add a small cost that encourages the sub-goal to be close to the current end-effector pose.

**Consistency**: Since the solver iteratively solves the problem at a high frequency and the noise from the perception pipeline may propagate to the solver, we find it useful to include a consistency cost that encourages the solution to be close to the previous solution.

**(Dual-Arm only) Self-Collision Avoidance**: To avoid two arms collide with each other, we compute the pairwise distance between the two point sets, each including the gripper points and grasped object points.

## A.9 Implementation Details of Path Solver

The path problems are implemented and solved using SciPy [128]. The number of decision variables is calculated based on the distance between the current end-effector pose and the target end-effector pose. Specifically, we define a fixed step size (20cm and 45 degree) and linearly approximate the desired number of "intermediate poses", which are used as decision variables. As in the sub-goal problem, they are similarly represented using position and Euler angles with the same bounds. For the first solving iteration, the initial guess is chosen to be linear interpolation between the start and the target. We similarly use sampling-based global optimization followed by a gradient-based local optimizer in the first iteration and only use local optimizer in subsequent iterations. After we obtain the solution, represented as a number of intermediate poses, we fit a spline using the current pose, the intermediate poses, and the target pose, which are then densely sampled to be executed by the robot.

In the objective function, we first unnormalize the decision variables and use piecewise linear interpolation to obtain a dense sequence of discrete poses to represent the path (referred to as "dense samples" below). A spline interpolation would be aligned with how we postprocess and execute the solution, but we find linear interpolation to be computationally more efficient. Below we discuss the individual cost terms in the objective function.

**Constraint Violation**: Similar to that in the sub-goal problem, we check violation of the ReKep constraints for each dense sample along the path and penalize with large weights.

**Scene Collision Avoidance**: The calculation is similar to the sub-goal problem, except that it is calculated for each dense sample. We ignore the collision calculation with a 5cm radius near the start and the target poses, as this tends to stabilize the solution when solved at a high frequency due to various real-world noises. We additionally add a table clearance cost that penalizes the path from penetrating the table (or the bottom of the workspace for the wheeled single-arm robot).

**Path Length**: We approximate the path length using the dense samples by taking the sum of their differences. Shorter paths are encouraged.

**Reachability**: We solve an IK problem for each intermediate pose inside the objective function as in the sub-goal problem. See the sub-goal solver section for more details.

**Consistency**: As in the sub-goal problem, we encourage the solution to be close to the previous one. Specifically, we store the dense samples from the previous iteration. To calculate the solution consistency, we use the pairwise distance between the two sequences (treated as two sets) as an efficient proxy. Alternatively, Hausdorff distance can be used.

**(Dual-Arm only) Self-Collision Avoidance**: We similarly compute self-collision avoidance for the dual-arm platform as in the sub-goal problem. We also use pairwise distance between the two sequences to efficiently calculate this cost.

### A.10  Comparisons with Prior Works on Visual Prompting for Manipulation

There has been several concurrent works investigating the application of visual prompting of VLMs to robotic manipulation [99–101, 112, 125]. Below we summarized the differences to highlight the contributions of this work.

**Task DoF**: In this work, we focus on challenging tasks that require 6 DoF (single arm) or 12 DoF (two arms) motions. However, this is not trivial for existing VLMs which operate on 2D images – as quoted from MOKA [100], "current VLMs are not capable of reliably predicting 6-DoF motions" and PIVOT [101], "generalizing to higher dimensional spaces such as rotation poses even additional challenges". To tackle this, one key insight from ReKep is that VLMs only need to implicitly specify full 3D rotations by reasoning about keypoints in *(x, y, z) Cartesian coordinates*. After this, actual 3D rotations are solved by high-precision and efficient numerical solvers, effectively sidestepping the challenge of explicitly predicting 3D rotations. As a result, the same formulation also naturally generalizes to controlling multiple arms.

**High-Level Planning**: While many works also consider multi-stage tasks via an language-based task planners which are independent from their methods, our formulation takes inspiration from TAMP and organically integrates high-level task planning with low-level actions in a unified continuous mathematical program. As a result, the method can naturally consider *geometric dependencies* across stages and do so *at a real-time frequency*. When a failure occurs, it would backtrack to a previous stage in which its conditions can still be satisfied. For example, in the "pouring tea" task, the robot can only start tilting the teapot when the teapot spout is aligned with the cup opening. However, if the cup is being moved in the process, it should level the teapot and re-align with the cup. Or if the teapot is being taken from the gripper, it should instead re-grasp the teapot.

**Low-Level Execution**: A common issue with using VLMs is that it is computationally expensive to run, hindering the high-frequency perception-action feedback loops often required for many manipulation tasks. As a result, most of existing works either consider the open-loop settings where visual perception is only used in the beginning or only consider the tasks where slow execution is acceptable. Instead, our formulation natively supports a high-frequency perception-action loops by coupling VLMs with a point tracker, which effectively enables reactive behaviors via closed-loop execution despite leveraging very large foundation models.

**Visual Prompting Methods**: We uniquely consider using visual prompting for code-generation, where code may contain arbitrary arithmetic operations on a set of keypoints via visual referring expressions. Although a single point is limiting to capture complex geometric structure, multiple points and their relations can even specify *vectors*, *surfaces*, *volumes*, and their *temporal dependencies*. While being conceptually simple, this offers a much higher degree of flexibility which can fully specify 6 DoF or even 12 DoF motions.

### A.11 Extended Discusssions on Limitations

Herein we present additional limitations of the existing system.

**Prompting and Robustness**: Although we have demonstrated that existing VLMs possess rudimentary capabilities at specifying ReKep constraints, we have observed that when dealing with tasks that span many stages with several temporally dependent constraints (A.14), the VLMs lack enough robustness to obtain consistent success.

**Task-Space Planning**: To enable efficient planning, in this work we only consider planning in the task space with the end-effector poses as decision variables. However, we have observed that in certain scenarios, it may be kinematically challenging for robots to achieve the optimized poses as the solver does not explicitly account for the kinematics of the robot. Planning in joint space can likely resolve the issue but we find it to be less computationally efficient for our tasks.

**Articulated Object Manipulation**: In this work, we do not investigate tasks involving articulated objects, as we observed this requires advanced spatial reasoning capabilities that are beyond those of existing VLMs. However, the ReKep formulation may be extended to such tasks by representing different types of joints also by "relations of keypoints". For example, ReKep constraints can be written to constrain certain keypoints to move only alongside a line (prismatic joints) or a curve (revolute joints). To extend to these scenarios, finetuning may be required as in [150–152].

**Bimanual Coordination**: Although we demonstrate the application of ReKep to bimanual manipulation, we also identify several important limitations in this domain. Notably, the challenges can be roughly categorized into those pertaining to semantic reasoning of keypoint relations by the VLM and those pertaining to solving for bimanual motions by the optimization solver. For semantic reasoning, to achieve bimanual folding, the VLM needs to possess certain spatial knowledge about which steps should/can be performed together by both arms. For example, the bottom of a shirt often needs to be grasped by two hands, each at one corner, in order to fold it upwards to align with the collar. Another example in blanket folding is to recognize that the bottom-left corner should be aligned with the top-left corner and the bottom-right corner should be aligned with the top-right corner, as other matching may lead to self-collision. For optimization solver, as bimanual motion planning dramatically increases the search space of possible motions, which slows down the overall pipeline and more frequently produces less optimal behaviors.

### A.12 Simulation Experiments

We additionally implement ReKep in OmniGibson [153] for the *Pour Tea* task. It is compared to a monolithic learning-based baseline based on the transformer architecture [154] adopted from RVT [155, 156]. The baseline is trained via imitation learning on 100 expert demonstrations, where demonstrations are from scripted policies using privileged simulation information. Success rates are averaged across 100 trials and reported below. Although the monolithic policy excels in training scenarios given its access to expert demonstrations, we observe that ReKep performs significantly stronger in unseen settings, and more importantly, without the need of expert demonstrations.

|                   | Seen Poses | Unseen Poses | Unseen Objects |
| ----------------- | ---------- | ------------ | -------------- |
| Monolithic Policy | **0.93**   | 0.31         | 0.14           |
| ReKep (Zero-Shot) | 0.75       | **0.68**     | **0.72**       |

### A.13 Comparisons of Visual Feature Extractors for Keypoint Proposal

Herein we provide qualitative comparisons of different methods for keypoint proposal. We compare three pre-trained visual feature extractors, each of which represents a popular class of pre-training methods: DINOv2 [5] (self-supervised pre-training), CLIP [94] (vision-language contrastive pre-training), and ViT [157] (supervised pre-training). We also compare to a variant that does not use

Segment Anything (SAM) [132] for its objectness prior. In Fig. 8, we show the extracted feature maps (projected to RGB space) and their clustered keypoints for three different scenes.

We would like to note two important observations from the comparisons: 1) objectness prior given by SAM is critical to constrain the keypoint proposal on objects in the scene instead of on the background, and 2) while most visual foundation models can provide useful guidance, DINOv2 produces sharper features that can better distinguish fine-grained regions of an object. The first observation can be clearly made by comparing the last column with other candidate methods. The second observation can be made by noting the following places: 1) the unique cyan color on the cup handle in the first scene, 2) the unique colors of the box panels in the second scene, and 3) the blue/green contrast of the top panel and the side panel in the third scene. Similarly, CLIP provides different features between different object parts, but the features are less sharp than those of DINOv2 (color saturating from one part to the other). ViT, on the other hand, produces least distinguishable features between object parts, especially when texture is similar. In general, our observations are aligned with other works that also apply DINOv2 for its fine-grained object understanding [114, 121, 123].

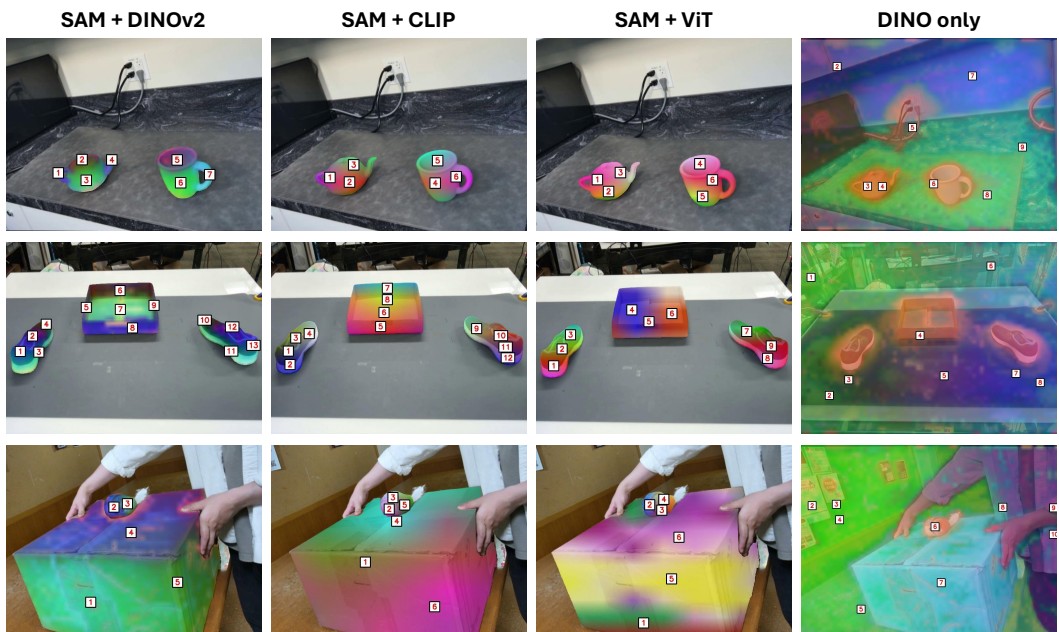

**Figure 8:** Comparisons of different methods for keypoint proposal.

## A.14 Case Study of Long-Horizon Tasks

To stress test the presented system, we additionally perform a case study of a long-horizon bimanual task of preparing breakfast tray. A successful completion requires 10 stages: 1) grasp table cloth, 2) place table cloth inside the tray, 3) grasp bread from the plate, 4) place bread on top of the table cloth, 5) grasp mug and teacup simultaneously with two arms, 6) align mug and teapot, 7) pour tea into the mug, 8) place the mug inside the tray, 9) grasp the two tray handles simultaneously with two arms, and 10) lift up the tray and hand it to the human operator.

This task presents significant challenges for many components in the system. Specifically, we find that existing pipeline for keypoint proposal and constraint specification are incapable of generating the full set of correct keypoints and the full sequence of the correct ReKep constraints. Additionally, we observe that as a result of multiple present objects, our point tracker is incapable of consistently tracking all required keypoints in the scene. As a result, we resort to manually annotated keypoints and constraints, as well as keypoint detection at only the start of each stage instead of persistent keypoint tracking. With the above modifications, we find that the system can reasonably perform the task. The initial and final configurations are shown below. The solutions for each stage are shown on the next page. The video result is available at rekep-robot.github.io.

Initial           Final

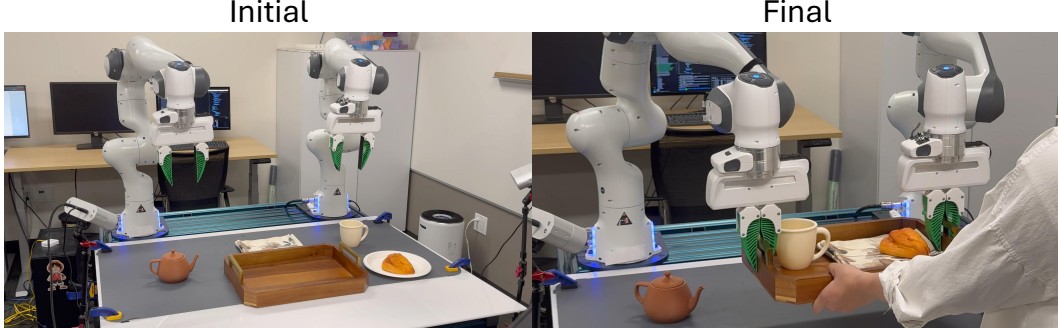

**Figure 9:** Initial configuration and successful final configuration of the preparing breakfast tray task.

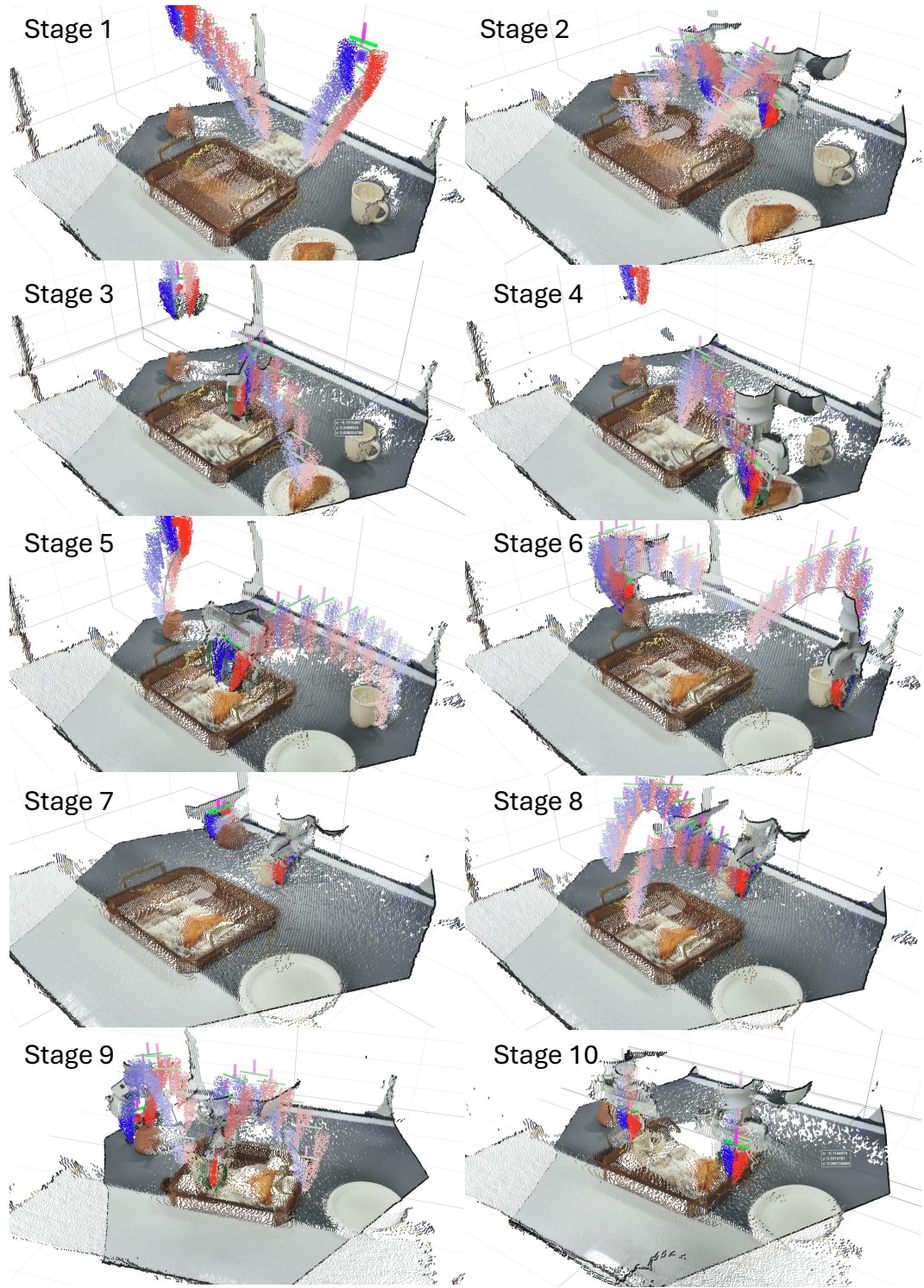

**Figure 10:** Solution visualization of the preparing breakfast tray task.

