# OpenReview forum: "ReKep: Spatio-Temporal Reasoning of Relational Keypoint Constraints for Robotic Manipulation"
_robot-learning.org/CoRL/2024/Conference — CoRL 2024_

### Official Review · Reviewer_JmTk · 2024-07-20
**Initial Review**

**Originality:** 2
**Technical Quality:** 2
**Clarity Of Presentation:** 4
**Potential Impact:** 2
**Recommendation:** 1
**Confidence:** 4

**Review:**

Quality: The proposed method is fairly intuitive and is mostly explained well, and the assumptions made are reasonable. There are a good number of demonstrations and visualizations of the method. Experiments address some important research questions, however the proposed model is only compared with a single baseline, and the number of experiment instances per task can be higher (perhaps in a simulated setting).

Clarity: The paper was written in an easily understandable manner. The figures provided are high quality and explanatory enough. However, it is not explicitly told in the paper how the framework determines the number of stages the task will have. This is better addressed in the explainer video (VLM decides the number of stages as well, alongside the constraints). I also did not understand how the system error breakdown was done. How do you quantify and compare errors from different modules (while considering their dependencies on each other)?

Originality: A fair number of previous works use VLMs and LVMs to generate keypoints or similar features to solve a manipulation problem (often, these features are either given to an optimizer as in this work, or the motion is also obtained from a learned model). Most of the proposed pipeline consists of off-the-shelf models. However, the method for generation of constraints as VLM-generated Python functions, although somewhat similar to VoxPoser, seems to be novel.

Significance: The significance of the paper is questionable because it’s claimed to provide a method that can take advantage of optimization solvers that can work with minimal human input. However, the prompt provided to the VLM contains overly detailed explanations of the tasks tested, which introduces a significant human bias and puts doubt on the actual capability of the VLM. The low amount of experiments also detract from the certainty of the results proposed.


Strengths:
- End-to-end framework: Given human instruction and RGBD images, the plan can be executed

- Shown to be suitable for online settings

- Robust to disturbances and failures, can replan from a previous stage

- Can generalize to (somewhat) new objects

- Shown to work with bimanual manipulation

Weaknesses:
- Overly detailed prompt that explains how the test tasks should be done in great detail, introducing a significant amount of human bias

- Most of the pipeline is a combination of readily available tools

- Only compared to a single baseline

- Said to be fully autonomous, however the non-annotated model achieves significantly worse results

- Not robust to a satisfactory level; as results in Table 2 show the non-annotated model only succeeds in about a fourth of the runs.

- In its current form, not feasible for complex manipulation problems (i.e., contact-rich), although this is discussed in the paper as well

- Heavy reliance on the performance of underlying VLM, LLM and point tracker, with the latter shown to be the largest contributor to system error. The existing model seems to use a single camera’s input, so this method could be extended to multi-view settings to reduce this error.

- More experiments could be done (possibly in a simulated environment) for a better analysis

**Quality Of The Limitations Section:**

2

**Questions For Rebuttal:**

- How would the proposed method perform with (much) less-informed instructions provided as an initial prompt?

- Current formulation uses ReKep costs in certain constraints only (i.e., requires them all to be <= 0). Is it possible to use these costs in other ways?

- Since LVM takes only the image input, the keypoints it generates can be unrelated to the task at hand, or even worse it may not generate some keypoints crucial to the task. If you have the language instructions anyway, why not use something like a VLM rather than a pure vision-based model?

- In your system error analysis, from what I understand you manually inspect failed experiments and select the first failing component as the cause of failure. However, this should naturally introduce a bias to yield higher error rates for earlier stages of the pipeline. Please clarify.

**Robotics Focus:**

4

**Summary Of Paper:**

This paper proposes an end-to-end pipeline to solve a manipulation problem given input RGBD images and a language description, with no additional inputs. It uses an LVM to generate keypoints for objects of interest. Input image with keypoints along with the language instruction are passed to a VLM to divide the task into stages and create cost functions for constraints in the form of simple Python functions. Finally, the constraints are passed to a motion optimizer to solve the motion planning problem. This framework can replan in case of failure, as well as generalize to novel objects and tasks. It is also shown to work with bimanual manipulation and potentially other multi-agent settings. The authors perform real-world experiments on a variety of settings such as pouring tea and folding clothes. They also provide some analysis of system error.

**Summary Of Recommendation:**

This paper proposes a method for language-instructed motion planning that is flexible and suitable for real-world applications. The explanation was mostly clear and intuitive, and visuals provided were sufficient and in high quality. For experiments, the authors provide a wide variety of hardware experiments that address several important research questions. However, the method is somewhat lacking in novelty; most of the pipeline consists of readily available tools with little modification or fine-tuning. The results of the experiments are not conclusive due to the low amount of experiment runs and comparison with only a single baseline which fails on almost every experiment task. Most important criticism is that the prompt provided to the VLM contains detailed explanations of many of the tasks used in the experiments, which raises questions about the actual capability of the VLM.

---

### Official Review · Reviewer_1ZB8 · 2024-07-21

**Originality:** 3
**Technical Quality:** 4
**Clarity Of Presentation:** 5
**Potential Impact:** 3
**Recommendation:** 3
**Confidence:** 5

**Review:**

Strengths:
- Novel system setup connecting VLMs and LLMs to verifiable, and interpretable low-level solvers for solving 3D problems
- Results show promising generalization of the methodology to several classes of problems
- Paper flows well with relevant coverage of literature review. However the formulation is a bit dense, and paper may benefit from simplifying its explanation further.
- The paper outlines an impressive systems effort in creating a full vertical of TAMP-like problem solving but while using much more generic VLMs and LLMs
- The hardware experiments are exciting in their scope to handle both rigid and deformable objects (i.e. clothes)
- The system may also be of great impact as an expert data generator, where non-modular specialized models may be trained on the data generated for complex 3D problems
- Appreciate the high throughput and reactive nature of solution

Weakness:
- Concerned about the hand-engineered nature of auxiliary constraint costs introduced in the system. The system is only evaluated on table-top manipulation setting. Even though the system does well on a range of such tasks, the costs rely heavily on the scoping of the problem space and I worry how this will generalize to non-tabletop manipulation or even harder mobile manipulation settings.
-  There are a number of confounding components in the system with no clear understanding of why this system works as well as it does. While the system will get better with more powerful VLMs and LLMs, I don't see a clear way of analyzing this problem more deeply to inform community of most promising gaps/bits of improvements. One clear systemic insight is the high failure rate of the 3D point tracker, but the reviewer still finds analysis lacking in depth.
- Missing comparisons with comparable non-modular baselines like a backbone-based end-to-end model solution. This will also open up the space to perform deeper analysis of what works/does not work/works better between MLLMs and more targeted backbone instantiation of this problem. Would appreciate more hardware/simulated experiments against baselines.
- Consider adding "Overall success" row to table in Figure 4. for a better understanding of performance

**Quality Of The Limitations Section:**

3

**Questions For Rebuttal:**

- What is the definition of "in-the-wild" in the context of task characteristics? Why is stowing a book a "in-the-wild" experiment?
- How will your system handle problems where contact is intentional and therefore the collision costs are variable? For example, compare against PerAct [1] with its explicit modeling of "collide" boolean which is off for intentional collisions in high-contact tasks.

[1] Shridhar, M., Manuelli, L. and Fox, D., 2023, March. Perceiver-actor: A multi-task transformer for robotic manipulation. In Conference on Robot Learning (pp. 785-799). PMLR.

**Robotics Focus:**

4

**Summary Of Paper:**

This paper proposes a novel scalable methodology to (a) break up a given task into stages using off-the-shelf LLMs and VLMs, (b) create constrained problem formulation for each stage such that, (c) off-the-shelf solvers can generate solutions good enough for real-time reactivity. The experiments consist of both single-arm and dual-arm configurations on intermediate-to-hard real-world problems, and show promising results.

**Summary Of Recommendation:**

I have concerns around complexity of the system and the unclear analysis/lack of tools for systemic analysis of gaps. That said, I find the scope of hardware results quite exciting and the system very impressive. I thus vote for a weak accept and look forward to the author's rebuttal.

---

### Official Review · Reviewer_v9TF · 2024-07-26
**Innovative approach; Some questions about the experiments**

**Originality:** 4
**Technical Quality:** 4
**Clarity Of Presentation:** 4
**Potential Impact:** 3
**Recommendation:** 3
**Confidence:** 4

**Review:**

The paper reads well, and I appreciate the real-world experiments. The concept of using relational keypoint constraints combined with an optimization step is particularly compelling. A key limiting factor appears to be the tracking of keypoints, which is acknowledged as a limitation by the authors.

However, some concerns remain. Details about the Voxposer baseline are missing, and the claim of handling "a large variety of manipulation tasks" seems a little bit overstated to me, as the experiments primarily focus on bimanual manipulation with compliant objects. Without some details about the VoxPoser baseline it makes it difficult for me to assess the relative performance of ReKep. Additionally, the experiments have a relatively short horizon, and the bimanual experiments are quite similar, e.g. similar clothes. It appears that the bimanual manipulation is achieved by executing the same method simultaneously with both arms, but the paper does not discuss the coordination between the two arms, which I see as a key challenge of bimanual manipulation. The term “Mobile arm” in this work and “mobile base” in Fig.6 seem to indicate mobile manipulation. I would prefer the terms “wheeled base” in Fig 6 or the term “single arm”.

Minor:

-	Line 36: combinatorially
-	Line 52 nonscientific language: ReKep enjoys …
-	Line 78: reprensetation
-	(Suggestion) Line 213ff: Instead of  a,b,c,d use m,w,b,r this makes the mapping easier. Alternatively, use a table with checkboxes.
-	Line 181: The symbol R has duplicate use in this work.
-	Video: The video is longer than the recommended length.
-	Video 0:45: Speed-up is missing for some experiments.
-	Fig2 shows a point cloud, but the text mention that keypoints are overlayed in the RGB image, but the Fig2 suggests that it’s the point cloud.
-	Fig3 is a little bit confusing. Could you overlay the waypoints in a camera image instead of a point cloud? The point cloud seems noisy,and I find it difficult to identify the waypoints.

**Quality Of The Limitations Section:**

2

**Questions For Rebuttal:**

- If the method works with arbitrary Python code. How can you ensure that the cost function is well specified. E.g. what happens if the output is -1 * np.linalg.norm(x, y) ?

- Could you do bimanual manipulation with rigid objects, e.g. a handover task or lifting an object?

- Why did you choose VoxPoser as baseline and how can you explain the relatively poor performance of VoxPoser? Is this Zero-Shot?

- Can you give more details on the additional challenges and how are those addressed?  “presenting additional challenges in bimanual coordination” (L. 249)?

- (Optional) Would a table in the related works section help to highlight the novelty of this work w.r.t. [80] and [81], [82].

- How is the bimanual coordination done? While querying the VLM is done for both robots at the same time, it is unclear to me how this step is executed. Are the two robots independently controlled independently or not?

**Robotics Focus:**

4

**Summary Of Paper:**

This work presents a VLM-based method for robotic manipulation tasks, named Relational Keypoint Constraints (ReKep). ReKep extracts keypoints from camera images using DINOv2 and interpolates them with segmentation masks. A Vision-Language Model (VLM), GPT4-o, is used to decompose the task into stages for high-level reasoning. The VLM is also queried to generate constraints and costs for each stage and keypoint. Costs are expressed with Python. An optimization solver then finds the end-effector pose that minimizes these costs. The method is evaluated in the real world with both a unimanual robot and a bimanual setup.

**Summary Of Recommendation:**

This is nice work. ReKep is an innovative approach with real-world experiments!

---

### Official Review · Reviewer_yq2j · 2024-07-31
**Reviews for the ReKep**

**Originality:** 3
**Technical Quality:** 3
**Clarity Of Presentation:** 3
**Potential Impact:** 3
**Recommendation:** 3
**Confidence:** 4

**Review:**

## Strengths:
1. One very interesting trial to combine VLM with classic numerical optimization methods. The optimization methods offer the additional numerical stability that is currently almost impossible for the VLM.
2. The ReKap pipeline does not require human labelling.
3. I found the implementation details in the appendix quite informative.
4. Real-wrold experiments are conducted to showcase the effectiveness of the ReKep.

## Weakness:
1. There are already some works[1,2] that utilized SoM + GPT4v to deal with the robotic tasks which makes this paper not that impressive.  Could the author make some comparisons among them?

2. Still limited to the tabletop manipulation tasks. Can this method be extended to the 3D ones, or harder ones, for the articulation tasks, as done in [3] where ChatGPT is used or in [4], where the model is fine-tuned?

3. Seems that the proposed system's path constrains mainly rely on cuRobo[5] library. Can the authors emphasize the contributions besides the usage of the cuRobo.

4. The baseline was only VoxPoser, more baselines are welcome.


[1] Huang, Haoxu, et al. "Copa: General robotic manipulation through spatial constraints of parts with foundation models." arXiv preprint arXiv:2403.08248 (2024).

[2] Hu, Yingdong, et al. "Look before you leap: Unveiling the power of gpt-4v in robotic vision-language planning." arXiv preprint arXiv:2311.17842 (2023).

[3] Xia, Wenke, et al. "Kinematic-aware Prompting for Generalizable Articulated Object Manipulation with LLMs." arXiv preprint arXiv:2311.02847 (2023).

[4] Huang, Siyuan, et al. "A3VLM: Actionable Articulation-Aware Vision Language Model." arXiv preprint arXiv:2406.07549 (2024).

[5] https://curobo.org/

**Quality Of The Limitations Section:**

2

**Questions For Rebuttal:**

1. See Review's Part.
2. The reason why you choose DINOv2 is not clear. Besides, you used the SAM with DINOv2 to get keypoints with the masks as the initial region, then why cannot we directly use DINOv2?
3. The error analysis. As mentioned in other API-calling methods like Instruct2Act, CaP, etc, the generated codes would contain syntax errors, how these errors can be handled? Or current GPT4o can already generate much better code lines?
4. Can you provide some rough token consumption information for each task execution?

**Robotics Focus:**

4

**Summary Of Paper:**

The authors proposes ReKep, which turns RGBD observations and language instructions into a series of constraints that map keypoints into numerical cost, based on which, optimized actions could be achieved in the action stage.

**Summary Of Recommendation:**

Good paper overal while some improvements is welcome.

---

### Author Rebuttal · Authors · 2024-08-12

We thank all the reviewers for their valuable feedback. We are glad to hear the overall positive remarks by the reviewers: *“one **very interesting trial** to combine VLM with classic numerical optimization methods”* (Reviewer yq2j), *“**Innovative approach**… the concept of using relational keypoint constraints combined with an optimization step is **particularly compelling**”* (Reviewer v9TF), *“a **novel scalable methodology** … connecting VLMs and LLMs to **verifiable**, and **interpretable** low-level solvers”* (Reviewer 1ZB8), and *”(demonstrated) in a **wide variety of hardware experiments**”* (Reviewer JmTk).

We are happy to report that we have included a new baseline in simulation as requested by the reviewers, an imitation learning policy learned on expert demonstrations. Our evaluations show that our proposed framework, despite requiring **no demonstrations or task-specific training**, performs significantly better under unseen settings. The results are reported below. The videos are included in the attached zip file.

|                   | Seen Poses | Unseen Poses | Unseen Objects |
|-------------------|------------|--------------|----------------|
| Monolithic Policy | **0.93**   | 0.31         | 0.14           |
| ReKep (Zero-Shot) | 0.75       | **0.68**     | **0.72**       |

We have also revised the paper and believe it is now much stronger thanks to reviewers’ suggestions. The new changes are marked in orange, and the revised PDF is included in the attached zip file.

For more details, please see the individual responses to each reviewer.

---

### Decision · Program_Chairs · 2024-09-04

**Decision:**

Accept

**Comment:**

This paper introduces Relational Keypoint Constraints (ReKep), a visually-grounded representation for constraints in robotic manipulation. Specifically, ReKep are expressed as Python functions mapping a set of 3D keypoints in the environment to a numerical cost. The system is implemented on a mobile single-arm platform and a stationary dual-arm platform that can perform a large variety of manipulation tasks. Most of the reviewers find this work interesting, while identifying several weaknesses that need to be addressed in the rebuttal. An important criticism is that the prompt provided to the VLM contains detailed explanations of many of the tasks used in the experiments, which raises questions about the actual capability of the VLM.
The reviews for this paper are mixed, with the majority of the reviewers recommending an accept. A reviewer raised the issue of the over-reliance on VLMs/LLMs without rigorous assessment, and the reliance on annotations. Other reviewers share these concerns, but do not seem to think they are strong enough for rejecting the paper.